# Diflubenzuron Induces Cardiotoxicity in Zebrafish Embryos

**DOI:** 10.3390/ijms231911932

**Published:** 2022-10-08

**Authors:** Xue Han, Xiaowen Xu, Tingting Yu, Meifeng Li, Yulong Liu, Jingli Lai, Huiling Mao, Chengyu Hu, Shanghong Wang

**Affiliations:** 1Department of Bioscience, School of Life Science, Nanchang University, Nanchang 330031, China; 2State Key Laboratory of Food Science and Technology, Nanchang University, Nanchang 330047, China

**Keywords:** diflubenzuron, zebrafish embryo, cardiotoxicity, oxidative stress, apoptosis

## Abstract

Diflubenzuron is an insecticide that serves as a chitin inhibitor to restrict the growth of many harmful larvae, including mosquito larvae, cotton bollworm and flies. The residue of diflubenzuron is often detected in aquaculture, but its potential toxicity to aquatic organisms is still obscure. In this study, zebrafish embryos (from 6 h to 96 h post-fertilization, hpf) were exposed to different concentrations of diflubenzuron (0, 0.5, 1.5, 2.5, 3.5 and 4.5 mg/L), and the morphologic changes, mortality rate, hatchability rate and average heart rate were calculated. Diflubenzuron exposure increased the distance between the venous sinus and bulbar artery (SV-BA), inhibited proliferation of myocardial cells and damaged vascular development. In addition, diflubenzuron exposure also induced contents of reactive oxygen species (ROS) and malondialdehyde (MDA) and inhibited the activity of antioxidants, including SOD (superoxide dismutase) and CAT (catalase). Moreover, acridine orange (AO) staining showed that diflubenzuron exposure increased the apoptotic cells in the heart. Q-PCR also indicated that diflubenzuron exposure promoted the expression of apoptosis-related genes (bax, bcl2, p53, caspase3 and caspase9). However, the expression of some heart-related genes were inhibited. The oxidative stress-induced apoptosis damaged the cardiac development of zebrafish embryos. Therefore, diflubenzuron exposure induced severe cardiotoxicity in zebrafish embryos. The results contribute to a more comprehensive understanding of the safety use of diflubenzuron.

## 1. Introduction

Diflubenzuron is mainly used as a highly effective insecticide in citrus, cotton, mushroom and ornamental plants; pastures, sewage systems and large areas of general outdoor treatment sites [1,2]. Diflubenzuron (DFB) is a kind of benzoyl urea insecticide that inhibits chitin exoskeleton formation and which has been applied for controlling agricultural pests [3]. In the toxicology analysis, the heart is often attacked by insecticides. Deltamethrin and pyridaben are widely used pyrethroid insecticides; both of them can also lead to serious cardiac abnormalities and dysfunction [4,5]. However, over-using always causes residue in water and threatens aquatic animals and even humans.

Diflubenzuron is also applied to kill the parasites of aquatic animals, which can cause residue in estuaries and oceans through the immoderate discharge of aquaculture sewage [1]. Diflubenzuron is stable in fishing grounds for a long time, with a half-life of up to 100 days [6]. Therefore, the toxicity of diflubenzuron to aquatic animals requires urgent exploration.

It has been reported that the residue of diflubenzuron in water easily causes acute and chronic damages to aquatic animals [7]. The tolerance level of marine and freshwater fishes to diflubenzuron are 100 mg/L and 50 mg/L, respectively [2,8]. Even more frightening is that the residue of diflubenzuron may transfer to humans through diet, which threatens human heath [9]. However, the toxicity and toxic mechanisms of diflubenzuron to organisms are still unknown. 

Zebrafish is a popular model organism for evaluating the toxicity of pollutants [10]. Zebrafish have the advantages of a highly conservative organ system, small size, transparent embryos, short developmental cycle and easy breeding [11,12]. There are about 30,000 genes in the zebrafish genome, which are similar to human genes with the conservation of 87% [13,14] Therefore, zebrafish are used as a model to reveal the mechanisms of diseases, and for drug screening and toxicology evaluation, etc. [15,16]. The heart is the first organ developed in zebrafish. After a series of complex developmental processes, the zebrafish embryo forms a mature organ that is capable of pumping blood [17,18]. The intact heart begins with the differentiation of myocardial and endocardial progenitors. The early heart includes the endocardium and myocardium, which form the atria and ventricles [11]. Any defects in the developmental process may lead to a dysfunction of heart and even death [19]. Therefore, damage to heart development is always regarded as a vital indicator in process of assessing pollutant toxicity. 

In this study, zebrafish embryos were used for evaluating the toxicity of diflubenzuron through indexes including mortality, morphological changes, the proliferation of myocardial cells, vascular development, oxidative stress and apoptosis.

## 2. Results

### 2.1. Developmental Toxicity of Zebrafish Embryos Is Induced by Diflubenzuron

Mortality was calculated after zebrafish embryos were separately exposed to diflubenzuron. The results showed that death of zebrafish embryos was increased at high concentrations of diflubenzuron exposure. The preliminary experiments showed no toxic phenotypes in zebrafish embryos when the concentration of diflubenzuron was lower than 0.5 mg/L, however a high concentration of diflubenzuron (over 3.5 mg/L) caused over 60% mortality (Appendix A). The lethal concentration 50 (LC_50_) of 72 hpf and 96 hpf were about 3.8 mg/L and 3.5 mg/L, respectively (Figure 1A). As the concentration of diflubenzuron increased, the heart rate of embryos gradually declined and hatchability was also reduced (Figure 1B,C); it also caused severe edema of the embryonic pericardium at high concentrations (Figure 1D). In addition, diflubenzuron exposure had no significant effects on the body length of zebrafish embryos (Figure 1E).

### 2.2. Diflubenzuron Exposure Causes the Cardiovascular Injure in Zebrafish Embryos

The decrease of heart beat and pericardium edema indicate that diflubenzuron exposure may induce the cardiotoxicity of zebrafish embryos. To further confirm the cardiotoxicity of zebrafish embryo induced by diflubenzuron, we carried out some experiments such as immunofluorescence, HE staining and fluorescence detection in heart and blood vessels of transgenic zebrafish model. The results showed that diflubenzuron exposure expanded the distance of SV-BA; the distance of SV-BA at the highest concentration was 1.59 times higher than that of the control group (Figure 2A,B). The results of HE staining also showed that diflubenzuron exposure caused the injures of sinus venous and bulbus arteriosus in zebrafish embryos (Appendix A). The results of immunofluorescence showed that diflubenzuron exposure inhibited the proliferation of cardiomyocytes (Appendix A). Furthermore, our results showed that diflubenzuron exposure caused damages to vasculature at the posterior region of zebrafish (Figure 2C).

### 2.3. Diflubenzuron Exposure Inhibits the Expression of Cardiac Development Genes

The toxicity of cardio development always affects the expression of the related genes, so genes such as vmhc, nppa, myh6, gata4 and tbx5 in the ventricle and atrium were detected in this study. The results showed that diflubenzuron exposure decreased the expression of these genes (Figure 3A–E). The expression of the genes in atrioventricular valve development (bmp4 and tbx2b) and endocardial conduction (klf2a) were also inhibited under diflubenzuron exposure (Figure 3F–H). In addition, the gene expression in the BMP signaling pathway (id1 and id2) and calcium-dependent signaling pathway (tnnc1a and atp2a1) were also abnormal after diflubenzuron exposure (Appendix A).

### 2.4. Diflubenzuron Exposure Induces Oxidative Stress in Zebrafish Embryos

The level of oxidative stress in zebrafish embryos was detected. Diflubenzuron exposure promoted the accumulation of reactive oxidative species (ROS) in the heart; the ROS intensity at the highest concentration was 2.4 times higher than that of the control group (Figure 4A,B). Moreover, the exposure of diflubenzuron increased the content of MDA and inhibited the activities of superoxide dismutase (SOD) and catalase (CAT) in a dose-dependent manner (Figure 4C–E). These results showed that diflubenzuron exposure activated oxidative stress in zebrafish embryos.

### 2.5. Diflubenzuron Exposure Leads to Heart Apoptosis 

To further study whether diflubenzuron-induced oxidative stress activates apoptosis, AO staining was performed in embryos. Diflubenzuron exposure led to increasing granular green spots (apoptotic cells) in the heart region (Figure 5A). Moreover, the expression of apoptosis-related genes was detected, including p53, bax, bcl2, caspase3 and caspase9. The results showed that the expression of apoptosis-related genes was up-regulated in the embryos exposed to diflubenzuron (Figure 5B–F).

## 3. Discussion

It is noteworthy that indiscriminate use of diflubenzuron often causes residue in the environment. The residue of diflubenzuron shows a high concentration in marine and freshwater [2,8]. Currently, there are few studies about the toxicity of diflubenzuron; especially, the specific target organs of diflubenzuron are still unknown. Therefore, it is necessary to explore the toxic effect of diflubenzuron. 

In this study, zebrafish embryos were selected to evaluate the toxicity of diflubenzuron. Our results show that exposure to diflubenzuron leads to the death of embryos, decreases heart beats rate and induces severe pericardial edema (Figure 1), which suggests that diflubenzuron exposure induces cardiac developmental toxicity of zebrafish embryos. 

Zebrafish embryo cannot survive a diflubenzuron concentration of 4.5mg/L. Therefore, the lethal concentration 50 (LC_50_) of diflubenzuron for zebrafish at 96 hpf was analyzed (3.5 mg/L), which is lower than those of other insecticides. For example, LC_50_ of bifenazate and thiophanate-methyl for zebrafish at 72 hpf was 14.5 mg/L [20] and 50.45 mg/L [21], respectively. Flupyradifurone is a new butenolide insecticide, and it was found that the LC_50_ of flupyradifurone to zebrafish embryos at 96 hpf is 210 mg/L [22]. Thus, among these insecticides, diflubenzuron shows a higher toxicity to zebrafish at the same concentration.

The early stage of heart development of zebrafish is followed by cardiogenic formation and differentiation, migration and the fusion of a bilateral heart, formation and spiral of myocardial tubes, cardiac circulation and ventricular ballooning [10]. The normal morphology of the ventricle and atrium are critical for zebrafish circulation. Triadimefon-induced pericardial edema increases the distance between the venosus and bulbus arteriosus, which leads to cardiovascular toxicity [23]. It is known that the malformation of the ventricle and atrium always influences the proliferation of myocardial cells and vascular development [24]. Transgenic Tg (*fli: GFP*) zebrafish are often used to analyze vascular development [25]. Proliferating cell nuclear antigen (PCNA) is an important regulator of the cell cycle, which is a typical marker for the proliferation of cells [26]. Diflubenzuron exposure also leads to a mechanical stretching of the heart, inhibits the proliferation of myocardial cells and damages the development of vascular system (Figure 2 and Appendix A). The abnormal cardiac looping caused by diflubenzuron may affect heart rate and blood flow. Therefore, diflubenzuron targets the development of the cardiovascular system in zebrafish. 

The developmental damage of the ventricle and atrium always leads to the abnormal expression of some related genes. The vmhc gene is a marker of the formation of ventricular myocardium during endocardiogenesis. The genes myh6 and vmhc encode the myosin heavy chains and act as cardiac development markers in activating the generation of ventricular cardiomyocytes [27,28]. Nppa is associated with myocardial development and the formation of atria and ventricles during the early stages of cardiac development [29]. During the whole embryonic and fetal development, the decrease of nppa expression may indicate that atrial and ventricular functions are affected [30,31]. Gata4 and tbx5 encode major transcription factors used in cardiac development. Deficiency or mutation of gata4 can lead to congenital heart injury [32]. Klf2a is a gene related to the cardiovascular system formation, which is a precursor marker of zebrafish heart valves [33]. Abnormal expression of tbx2b and bmp4 inhibits the formation of atrioventricular tube and destroys cardiac cyclization [34,35]. Exposure of diflubenzuron inhibits these genes’ expression, which indicates that the development of the ventricle and atrium is severely impaired in zebrafish embryos (Figure 3).

The BMP signaling pathway and calcium-dependent signaling pathway play critical roles in regulating cardiovascular development [36,37,38]. Id1 and id2 are members of the BMP signaling pathway. Knockout of Id1 or Id2 causes severe heart damage [39]. Tnnc1a and atp2a1 are regulators for the calcium channel. Tnnc1a controls the function of cardiac contraction through regulating the activity of troponin C [40]. In addition, atp2a1 is an ATPase-related protein, which can coordinate the transport of ATPase and Ca^2+^ [38]. The damage of the BMP signaling pathway and calcium-dependent signaling pathway by diflubenzuron exposure likely inhibits cardiac development of zebrafish embryos (Appendix A).

The abnormal expression of development-related genes is always accompanied by the activation of oxidative stress [40]. Oxidative stress is induced by the imbalance between the production of reactive oxygen species (ROS) and the antioxidant defense system, which is an important index to evaluate drug toxicity [41]. The homeostasis of cells is regulated by antioxidants and oxidants. SOD catalyzes the transformation of superoxide anion radical into H_2_O_2_ [42]. CAT is responsible for the subsequent degradation of H_2_O_2_ to produce non-toxic H_2_O [43,44,45]. MDA content usually reflects the level of lipid peroxidation, which is considered as the main cause of impaired cell function caused by reactive oxygen species overexpression [46,47]. The pollutant always induces oxidative damages to body. Exposure to prothioconazole causes lipid peroxidation and oxidative damage in zebrafish embryos [48]. As expected, diflubenzuron exposure increased the content of ROS and MDA and inhibited the activities of SOD and CAT, which suggested that oxidative stress of embryos was activated (Figure 4). 

Oxidative stress is one of the toxic mechanisms caused by the entry of most external compounds into the organism. It destroys the immune function of the organism, causes damage to cell components and even leads to cell apoptosis in serious cases [40,49]. Oxidative stress induced by combined exposure of triazophos and imidacloprid to zebrafish embryos may be the basis of the mechanism of the apoptosis effect [50]. AO staining showed that the number of apoptotic cells in heart region increased under diflubenzuron exposure (Figure 5A). Moreover, the expression of apoptosis-related genes (bax, p53, caspase3 and caspase9) was also activated (Figure 5B–F). Therefore, we speculate that the developmental toxicity and neurotoxicity induced by diflubenzuron may be caused by ROS-induced apoptosis in the process of proliferation. 

In summary, this study provides valuable knowledge on diflubenzuron leading to cardiotoxicity in zebrafish embryo. Thus, diflubenzuron may threaten other aquatic animals. We should strictly follow the instructions and strengthen the awareness of safety protection when using diflubenzuron.

## 4. Materials and Methods

### 4.1. Zebrafish Husbandry and Experimental Reagents

Tg (*cmlc2: GFP*) transgenic line, Tg (*fli: GFP*) transgenic line and wild-type (AB strain) zebrafish were purchased from Nanjing EzeRinka Biotechnology Co., Ltd. (Nanjing, China) and fed with newly hatched brine shrimp two times per day. Tg (*cmlc2: GFP*) transgenic zebrafish embryos were used to study heart development. Tg (*fli: GFP*) zebrafish transgenic zebrafish embryos were used to analyze vascular development. The Tg (*cmlc2:GFP*) zebrafish embryos were exposed to concentrations of 0.5, 1.5 and 2.5 mg/L of diflubenzuron from 6 hpf to 96 hpf. The zebrafish living environment was a photoperiod (light for 14 h, dark for 10 h) at 28.5 °C. The circulating water system maintains normal operation, natural conductivity (500–550 µs/cm) and pH (7.0–7.4). The fertilized embryos were obtained from natural spawning of adult zebrafish and incubated in embryo culture medium (5 mM NaCl, 0.17 mM KCl, 0.33 mM CaCl_2_, 0.33 mM MgSO_4_, pH 7.4) at 28 °C.

Diflubenzuron (CAS: 35367-38-5, analytical standard) was bought from Aladdin (Shanghai, China). Dimethyl sulfoxide (DMSO) (Sigma, St. Louis, MO, USA) was used to prepare the stock solutions of diflubenzuron. N-phenylthiourea (PTU) (Sigma, St. Louis, MO, USA) was used to inhibit the production of embryonic melanin. The kits of superoxide dismutase (SOD), catalase (CAT), malondialdehyde (MDA) and reactive oxygen (ROS) were bought from Nanjing Jiancheng Bioengineering Institute. The protein concentration was determined by the total protein assay kit (Nanjing, China). 

### 4.2. Diflubenzuron Exposure and Morphological Changes of Zebrafish Embryos

Based on OECD guidelines, 5-h post fertilization (hpf) embryos were collected and transferred to 6-well plates with 30 embryos per well. A stock solution (1 mg/L) was prepared by dissolution of diflubenzuron powder in DMSO, and stored at 4 °C. We performed relevant preliminary experiments of diflubenzuron exposure, including concentration below 0.5 mg/L and above 4.5 mg/L.

They were separately exposed to the different concentrations of diflubenzuron in 6-well plates (20 mL of embryo culture medium per well), including 0.5 mg/L, 1.5 mg/L, 2.5 mg/L, 3.5 mg/L and 4.5 mg/L. 0.01% of DMSO (containing 20 µL of DMSO in 20 mL of embryo culture medium) was used as a control. After exposure, the embryos were collected and their mortality and hatchability were calculated at 96 hpf. A stereomicroscope (SMZ800N-FL) was used to observe and manually measure pericardial area, heart rate (average beats per min) and the distance between the sinus vein and bulbus arteriolar (SV-BA) (μm). 24 embryos of each concentration were measured.

### 4.3. Real-Time Quantitative PCR

Gene transcription of *nppa*, *gata4*, *vmhc*, *tbx5*, *myh6*, *tbx2b*, *klf2a* and *bmp4* were detected at 96 hpf in zebrafish embryos exposed to various concentrations of diflubenzuron (0, 0.5, 1.5 and 2.5 mg/L). Total RNA was extracted from 60 embryos in each treatment group using RZ buffer (Tiangen, Beijing, China) at 96 hpf. A quick drop spectrophotometer (Molecular Devices, Sunnyvale, CA, USA) was used to determine the concentration and purity of total RNA with a target of OD_260_: OD_280_≈2.0. The first stage of cDNA synthesis was performed on 1 μg RNA using the QuantScript RT kit (TaKaRa, Kyoto, Japan). The qPCR was performed in CFX ConnectTM Real-Time System (Bio-Rad, Hercules, CA, USA) and its program was 95 °C for 5 min, followed by 40 cycles at 95 °C for 30 s, and 55 °C for 30 s. *β-actin* was used as the internal reference. The relative expression level of different genes in zebrafish embryos was calculated by 2^−^^△△CT^ method. The primer sequences are shown in Appendix A.

### 4.4. Hemotoxin and Eosin (HE) Staining

Heart sections in control and 0.5, 1.5 and 2.5 mg/L diflubenzuron-exposed were stained with H&E in zebrafish embryos at 96 hpf. Diflubenzuron-exposed embryos (96 hpf) were fixed and incubated overnight in 4% paraformaldehyde (PFA), and then washed 3 times with embryos culture medium. The embryos were dehydrated by ethanol gradient (70%, 80%, 100%). The fixed embryos were embedded in paraffin wax and cut into slices with a Leica microtome. A hematoxylin and eosin staining kit (Beyotime, Shanghai, China) was used and the protocol of the manufacturers was followed. Images were taken with an upright metallurgical microscope (Nikon, Tokyo, Japan). 

### 4.5. Oxidative Stress Analysis

A total of 24 embryos from each group were collected and the protein was harvested. The protein concentration was determined using a total protein assay kit (Nanjing, China). The embryos were exposed to diflubenzuron for 96 h. The activities of superoxide dismutase (SOD), catalase (CAT) and malondialdehyde (MDA) were calculated by SpectraMax^®^ i3x (Molecular Devices, Sunnyvale, CA, USA). In addition, the embryos were dyed with 10 μM of DCFH-DA (2,7-Dichlorofluorescein Diacetate) at 37 °C for 1 h under dark condition. The images were captured by a fluorescence stereomicroscope (Nikon, Tokyo, Japan). ROS fluorescence intensity was measured by Image J software.

### 4.6. Acridine Orange (AO) Staining

Diflubenzuron-exposed embryos (96 hpf) were collected and washed three times with embryo culture medium. The embryos were incubated with the culture medium containing 5 mg/L AO in darkness at 37 °C for 30 min. Then, the embryos were anesthetized and imaged by a fluorescent stereomicroscope (Nikon, Tokyo, Japan). Bright and granular green spots indicate the apoptotic cells in zebrafish embryos. 

### 4.7. Immunofluorescence

The Tg (fli: GFP) zebrafish embryos were exposed to concentrations of 0.5, 1.5 and 2.5 mg/L diflubenzuron at 96 hpf. Diflubenzuron-exposed embryos were collected in centrifuge tubes, then fixed with PFA overnight and washed 3 times with embryo culture medium. The embryos were dehydrated by ethanol gradient (70%, 80%, 100%). The fixed embryos were embedded in paraffin wax and cut into slices using a Leica microtome. Each of section was dewaxed, rehydrated and immersed in EDTA. The sections were blocked using 3% BSA blocking solution for 2 h. Then, the sections were incubated with mouse anti-PCNA antibody (Servicebio, Shenzhen, China) at 4 °C overnight. The sections were dyed with goat anti-mouse red fluorescence antibody of Cy3 (ZSGB-BIO, Beijing, China) for 2 h at room temperature. DAPI (Sangon, Shanghai, China) was used to stain nucleus. The images were taken with an upright metallurgical microscope (Nikon, Tokyo, Japan). 

### 4.8. Statistical Analysis

The data was analyzed with GraphPad Prism8.0 software. The values were expressed as mean ± SD. The significant difference was determined by one-way ANOVA followed by Tukey’s multiple comparison tests (* *p* < 0.05, ** *p* < 0.01 and *** *p* < 0.001). The fluorescence intensity was calculated by Image J software.

## Figures and Tables

**Figure 1 ijms-23-11932-f001:**
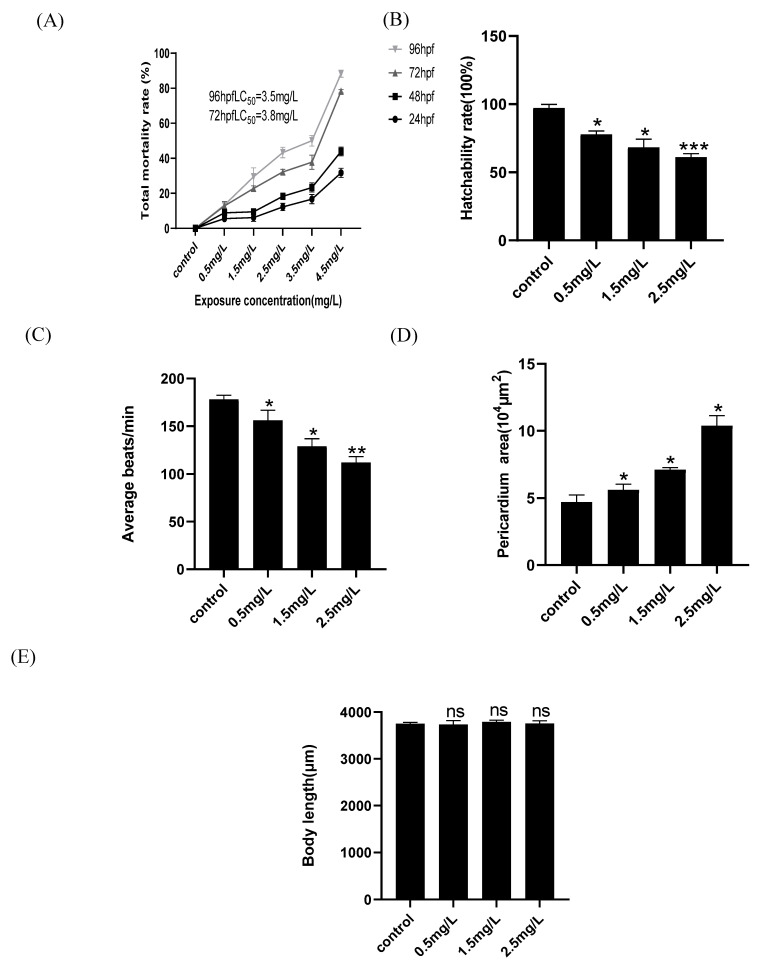
**Diflubenzuron induces zebrafish embryonic developmental toxicity.** Mortality of larvae after diflubenzuron exposure (**A**). The influence of diferent concentration of diflubenzuron exposure on hatchability rate (**B**), The qualification of heart rates of zebrafish larvae exposed to diflubenzuron (**C**), The qualification of pericardium area of zebrafish larvae exposed to diflubenzuron (**D**), The body length of zebrafish larvae exposed to diflubenzuron (**E**). Each bar represents mean ± SD (*n* = 24) of three independent experiments. The data represent mean ± SD, ns: no significant difference, * *p* < 0.05, ** *p* < 0.01, *** *p* < 0.001.

**Figure 2 ijms-23-11932-f002:**
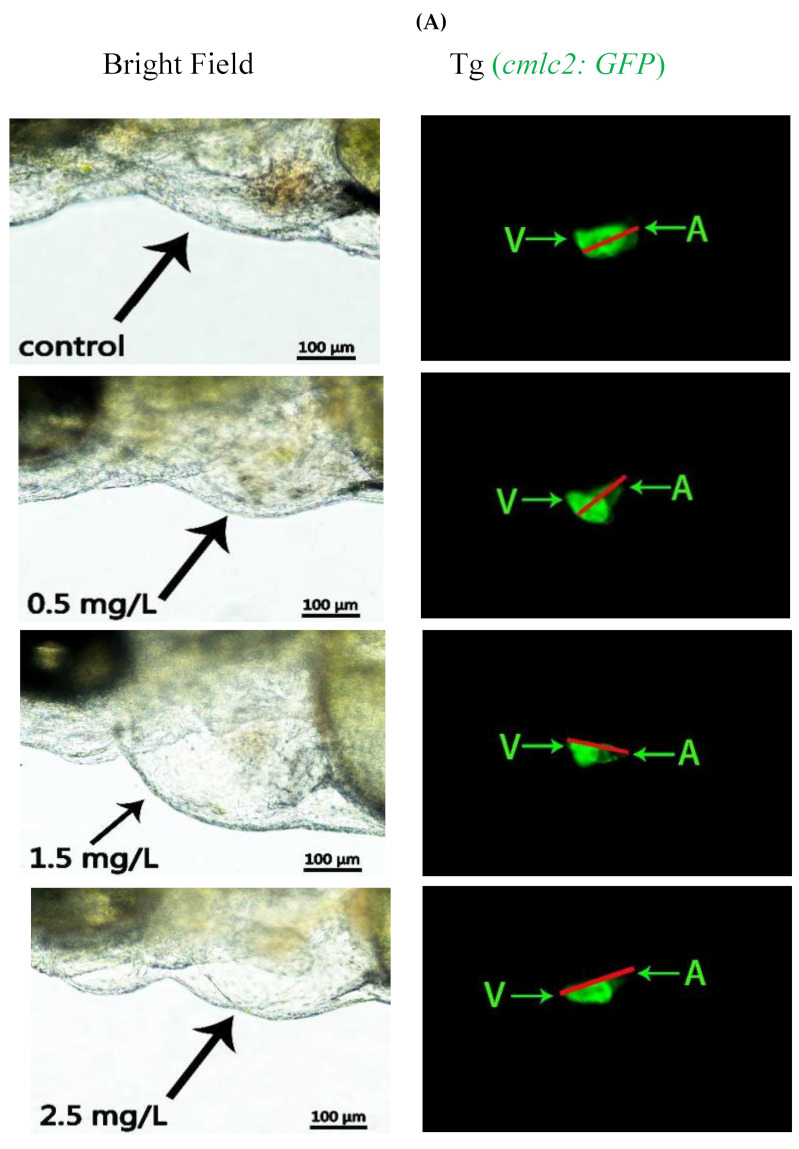
**Exposure to diflubenzuron induces cardiac developmental toxicity in zebrafish embryos.** The *Tg (cmlc2:GFP)* zebrafish embryos were exposed to diflubenzuron from 6 hpf to 96 hpf. (**A**), the arrows point to the heart. The qualification of the SV-BA distance of zebrafish larvae exposed to diflubenzuron at 96 hpf. (**B**), (*n* = 24, * *p* < 0.05, ** *p* < 0.01., mean ± SD) of three independent experiments. SV, sinus venous; BA, bulbus arteriosus. Scale bar: 500 μm., the arrows point to the ventricles and atria. The zebrafish embryos were exposed to 2.5 mg/L of diflubenzuron at 96 hpf. The red boxes indicated defective region of vascular system (**C**). Each of experiment was repeated at least three times.

**Figure 3 ijms-23-11932-f003:**
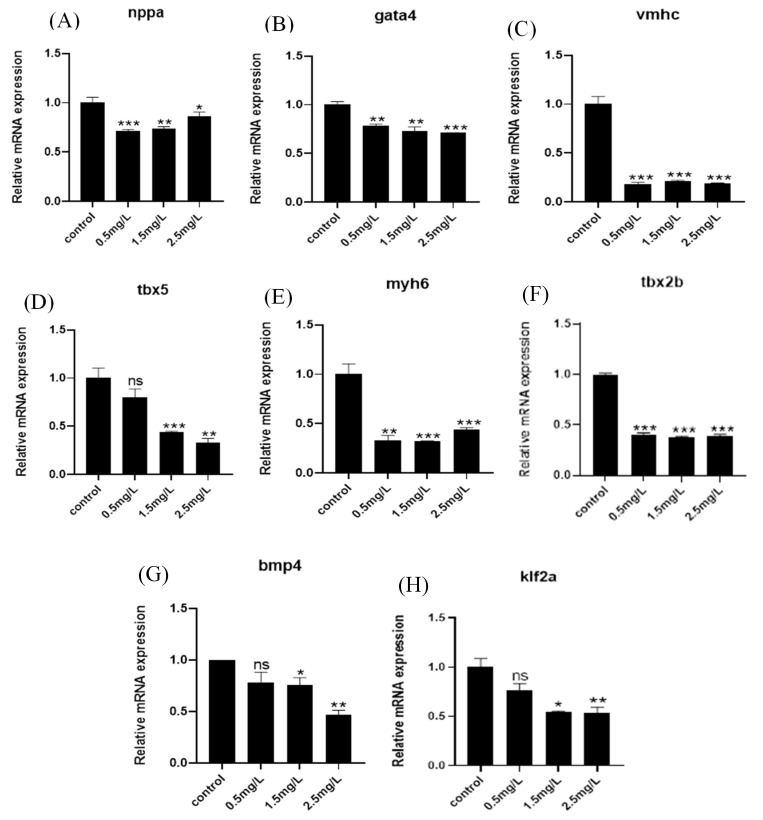
**Effects of diflubenzuron exposure on mRNA expression in genes related to heart development****.** Gene transcription of *nppa*, *gata4*, *vmhc*, *tbx5*, *myh6*, *tbx2b*, *klf2a* and *bmp4* in zebrafish embryos after exposure to diflubenzuron at 96 hpf. (**A**–**H**). Each treatment was repeated three times. Univariate ANOVA followed by Tukey test was used for differences between treatment groups. Different letters on the column indicate significant differences between treatments, ns: no significant difference, * *p* < 0.05, ** *p* < 0.01, *** *p* < 0.001 (Mean ± SD).

**Figure 4 ijms-23-11932-f004:**
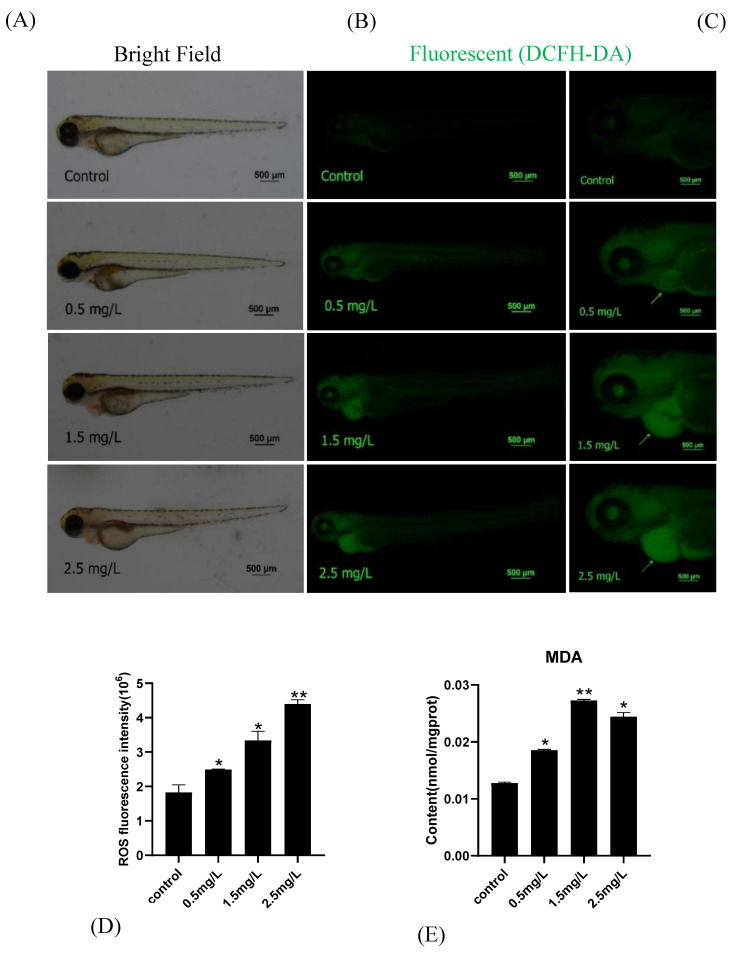
**Effects of diflubenzuron exposure on oxidative stress****.** ROS staining (**A**), Relative fluorescence intensity in heart region (*n* = 24) (**B**). the arrows point to the heart. The concentration of SOD, CAT and MDA were detected (**C**–**E**). Each bar represents mean ± SD (*n* = 24) of three independent experiments. Asterisks indicate significant differences from control, * *p* < 0.05, ** *p* < 0.01, *** *p* < 0.001.

**Figure 5 ijms-23-11932-f005:**
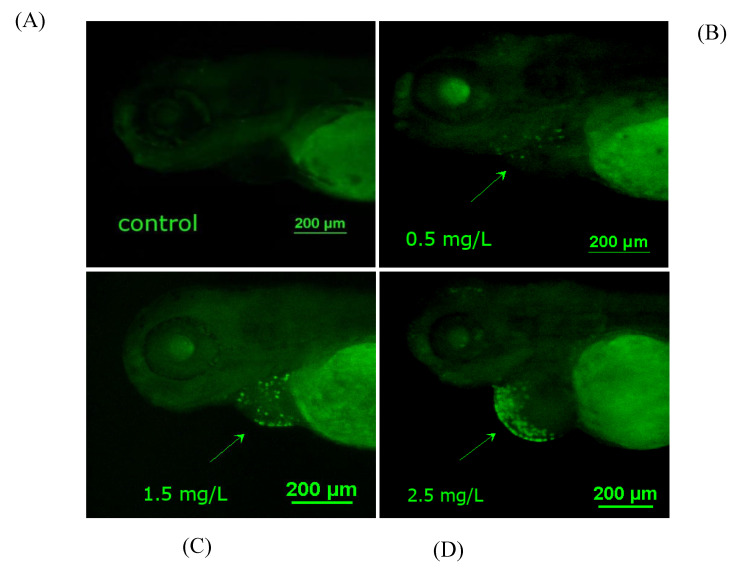
**Effects of diflubenzuron exposure on cardiac apoptosis in zebrafish embryos****.** Apoptotic cells of zebrafish induced by diflubenzuron exposure were detected by the AO staining at 96 hpf (**A**). Bright green dots represent apoptotic cells, and the arrows point to the heart. qRT-PCR was used to detect the expression of *bax*, *bcl2*, *p53*, *caspase3* and *caspase9* (**B**–**F**). Each bar represents mean ± SD (*n* = 24) of three independent experiments. Asterisks indicate significant differences from control, ns: no significant difference, * *p* < 0.05, ** *p* < 0.01, *** *p* < 0.001.

## Data Availability

Not applicable.

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
