# Peer review of "Diflubenzuron Induces Cardiotoxicity in Zebrafish Embryos"

_ijms, 2022, doi:10.3390/ijms231911932_

Round 1

Reviewer 1 Report (New Reviewer)

Abstract :

The sentence “In this study, diflubenzuron exposure showed severe cardiotoxicity to zebrafish embryos.” Is more like a conclusion and would be better placed just before the last sentence of the abstract.

If you describe SV-BA and ROS abbreviation you have to do the same for the other one (ex: SOD, MDA, CAT, AO) but abbreviation signification is not an obligation in the abstract. Try to be homogenous.

Introduction

The introduction can be is short and be improved (more exemple of effect on other species, exemple of other insecticide which have cardiac effect,….). However the aims of the study is clear and the introduction well organized

Correction:

·         The first paragraph lacks references for me. Please add at least one.

·         The last sentence “Generally, difluben-zuron exposure causes severe cardiotoxicity to zebrafish embryo” is misplaced. It is either a concluding sentence that gives the results of this study and therefore should be in the conclusion. Or it is to mention that Diflubenzuron is known to cause heart problems and in this case it should be mentioned above with references.

Results

The results presented here are very interesting. In general the result part needs to be reworked for the form.  The number of figures is huge even if grouped together to be limited "officially to 5". A big sorting can be done to transfer part of them in supplementary material/data or to present them in table form.

Corrections:

2.1

·         “Zebrafish embryos were separately exposed to diflubenzuron at different concentra-tion gradients (0.5 mg/L, 1.5 mg/L, 2.5 mg/L, 3.5 mg/L and 4.5 mg/L) (Fig. 1A)”. These informations would be more appropriate in the Materials and Methods section. Figure 1A is not necessary for me. How do you expose the embryos separately if they are all in a petri dish? The individual exposure of the fish can be done in a multi-well plate. Maybe you meant that each concentration was treated separately but in that case change the organization of the sentence to make it less confusing for the reader.

·         If you shown the figure 1F you have to add a least a mention of it in the text : Diflubenzuron have no significant effect on body length on ZF embryo at tested concentration (Fig. 1F).

·         Figure 1 : An effort can be done on the quality of the figure and the stat star (*) can have a larger size and be placed further away from the error bars to make the reading more pleasant for the reader.

2.2

·         “The result showed that diflubenzuron exposure ex-panded the distance of SV-BA (Fig. 2A and 2B). The same result was also obtained from HE staining (Fig. 2C).” maybe you can described a bit more your results

·         Figure 2: This figure have many quality problems (see below). In general, there is, for me, too much image (23) and information for one figure (Histograme+ picture+ histo+ fluo). If as mentioned above the figure, some results show similar results they can be put in additional material. Maybe you can only show the picture of control and higher concentration?.

·         Figure 2A: The images are blurred. On the pictures on the left we don't know what the arrow is pointing at. You can enlarge it to target the heart more precisely. On the left pictures, the red line is not very visible (not thick enough?)

·         Is Figure 2B a copy and paste of something? We see a "Fig. 2" which hides the legend of the y-axis.

·         On figure 2C and D the captions on the photos can be reworked to be more readable (bigger, change of color to make it more visible ...)

·         Figure 2E the images should be aligned and of the same size. The beginning of the legend is visible under the photos.

·         Legend : Some information can be transferred in M&M

2.3, 2.4, 2.5  :

·         Same that for figure 1: An effort can be done on the quality of the figure 3 and 4 and the statistic star (*) can have a larger size and be placed further away from the error bars to make the reading more pleasant for the reader. The quality of the picture can be largely improved.  Information missed in the legend (Mean +- SD? SE?)

·         You can you can better describe the results presented and give values or at least factors of increase (e.g. ROS intensity is significantly 2.5 times higher at the highest concentration than for the controls). Same for all histogram. For point 2.4 what is the purpose of figure 4A? it is not mentioned anywhere

Discussion

The discussion is relatively well constructed and pleasant to read.

Correction :

·         P14 : “ Diflubenzuron (DFB) is a kind of benzoyl urea insecticide that inhibits chitin exoskel-eton formation and has been applied for controlling agricultural pest [19]. ……. of them can also lead to serious cardiac abnormalities and dysfunction [20, 21].”  Can be transferred to the introduction part

·         P14 : “There-fore, the lethal concentration 50 (LC50) of diflubenzuron for zebrafish at 96 hpf was ana-lyzed (3.5 mg/L), which is lower than those of other insecticides. “ : insecticide with the same mode of actions?

·         More comparison with other pollutant/pesticide can be add to increase the strength of the arguments

·          

Material and method

No comments except the % of DMSO used to dissolve the diflubenzuron is missing and very important according to OECD guidelines.

Author Response

Abstract :

The sentence “In this study, diflubenzuron exposure showed severe cardiotoxicity to zebrafish embryos.” Is more like a conclusion and would be better placed just before the last sentence of the abstract.

Response: Thank you. We have placed this sentence to the last sentence of the abstract (lines of 33-34)

If you describe SV-BA and ROS abbreviation you have to do the same for the other one (ex: SOD, MDA, CAT, AO) but abbreviation signification is not an obligation in the abstract. Try to be homogenous.

 Response: Thank you. We have provided the abbreviations of SOD, MDA, CAT and AO in the abstract. 

Introduction

The introduction can be is short and be improved (more exemple of effect on other species, exemple of other insecticide which have cardiac effect,….). However the aims of the study is clear and the introduction well organized

Response: Thank you for your suggestion. We have removed some examples of other species, and refined the introduction.

Correction:

  • The first paragraph lacks references for me. Please add at least one.

Response: Thank you. We have added some relevant references in first paragraph of introduction (line 42).

  • The last sentence “Generally, diflubenzuron exposure causes severe cardiotoxicity to zebrafish embryo” is misplaced. It is either a concluding sentence that gives the results of this study and therefore should be in the conclusion. Or it is to mention that Diflubenzuron is known to cause heart problems and in this cause it should be mentioned above with references.

Response: Thank you for your suggestion. This sentence is the conclusion of our article, so, we have removed this sentence from the last paragraph of introduction (lines 33-34)

Results

The results presented here are very interesting. In general the result part needs to be reworked for the form.  The number of figures is huge even if grouped together to be limited "officially to 5". A big sorting can be done to transfer part of them in supplementary material/data or to present them in table form.

Response: Thank you. We have adjusted our figures according to your suggestion.

Corrections:

2.1

  • “Zebrafish embryos were separately exposed to diflubenzuron at different concentra-tion gradients (0.5 mg/L, 1.5 mg/L, 2.5 mg/L, 3.5 mg/L and 4.5 mg/L) (Fig. 1A)”. These informations would be more appropriate in the Materials and Methods section. Figure 1A is not necessary for me. How do you expose the embryos separately if they are all in a petri dish? The individual exposure of the fish can be done in a multi-well plate. Maybe you meant that each concentration was treated separately but in that case change the organization of the sentence to make it less confusing for the reader.

Response: Thank you. The sentence “Zebrafish embryos were separately exposed to diflubenzuron at different concentra-tion gradients (0.5 mg/L, 1.5 mg/L, 2.5 mg/L, 3.5 mg/L and 4.5 mg/L)” has been deleted. In addition, we also have deleted figure 1A. 

  • If you shown the figure 1F you have to add a least a mention of it in the text : Diflubenzuron have no significant effect on body length on ZF embryo at tested concentration (Fig. 1F).

Response: Thank you. We have added description of new fig 1E in lines of 84-86.

  • Figure 1 : An effort can be done on the quality of the figure and the stat star (*) can have a larger size and be placed further away from the error bars to make the reading more pleasant for the reader.

Response: Thank you. We have enlarged the stars in all figures.

2.2

  • “The result showed that diflubenzuron exposure ex-panded the distance of SV-BA (Fig. 2A and 2B). The same result was also obtained from HE staining (Fig. 2C).” maybe you can described a bit more your results

Response: Thank you. We have added the description about the results of HE staining in lines of 96-99.

  • Figure 2: This figure have many quality problems (see below). In general, there is, for me, too much image (23) and information for one figure (Histograme+ picture+ histo+ fluo). If as mentioned above the figure, some results show similar results they can be put in additional material. Maybe you can only show the picture of control and higher concentration?.

Response: Thank you for your suggestion. We have placed fig 2C and 2D into supplementary materials (served as a Fig. S2).

  • Figure 2A: The images are blurred. On the pictures on the left we don't know what the arrow is pointing at. You can enlarge it to target the heart more precisely. On the left pictures, the red line is not very visible (not thick enough?)

Response: Thank you. We have enlarged figure 2A. Due to the resolution of our stereomicroscope, we have tried our best capturing clear images. 

  • Is Figure 2B a copy and paste of something? We see a "Fig. 2" which hides the legend of the y-axis.

Response: I am sorry for our mistakes. We have adjusted figure 2B in a better format.

  • On figure 2C and D the captions on the photos can be reworked to be more readable (bigger, change of color to make it more visible ...)

Response: Thank you. We have reworked the captions of figure 2C and 2D,You can see in the supplementary material Fig. S2.

  • Figure 2E the images should be aligned and of the same size. The beginning of the legend is visible under the photos.

Response: Thank you. We have adjusted fig 2E for the same size

  • Legend : Some information can be transferred in M&M

Response: Thank you for your suggestion. We have transferred some information of legend to M&M.

2.3, 2.4, 2.5  :

  • Same that for figure 1: An effort can be done on the quality of the figure 3 and 4 and the statistic star (*) can have a larger size and be placed further away from the error bars to make the reading more pleasant for the reader. The quality of the picture can be largely improved.  Information missed in the legend (Mean +- SD? SE?)

Response: Thank you. We have revised figure 3 and 4. The missed information of legend was also added.

  • You can you can better describe the results presented and give values or at least factors of increase (e.g. ROS intensity is significantly 2.5 times higher at the highest concentration than for the controls). Same for all histogram. For point 2.4 what is the purpose of figure 4A? it is not mentioned anywhere

Response: Thank you. We have added detailed description for all histogram. In addition, the purpose of figure 4A was to detect ROS intensity using DCFH-DA probes, we have added relevant information in lines of 118-120.

Discussion

The discussion is relatively well constructed and pleasant to read.

Response: Thank you for your comments.

Correction :

  • P14 : “ Diflubenzuron (DFB) is a kind of benzoyl urea insecticide that inhibits chitin exoskel-eton formation and has been applied for controlling agricultural pest [19]. ……. of them can also lead to serious cardiac abnormalities and dysfunction [20, 21].”  Can be transferred to the introduction part

Response: Thank you. We have transferred these sentences to introduction part (lines of 42-46).

  • P14 : “There-fore, the lethal concentration 50 (LC50) of diflubenzuron for zebrafish at 96 hpf was ana-lyzed (3.5 mg/L), which is lower than those of other insecticides. “ : insecticide with the same mode of actions?
  • More comparison with other pollutant/pesticide can be add to increase the strength of the arguments

Response: Thank you. We have added more examples about the toxicity of insecticides and discussed the comparison between diflubenzuron with these other insecticides in lines of 147-149.

Material and method

No comments except the % of DMSO used to dissolve the diflubenzuron is missing and very important according to OECD guidelines.

Response: Thank you. The detailed description about the usage of DMSO was provided in line 249-250.

Reviewer 2 Report (New Reviewer)

the present research paper well revised  and properly organized,hence i recommend the paper for publication

Author Response

Response: Thank you for your evaluation.

Reviewer 3 Report (New Reviewer)

The manuscript study the cardiotoxicity induced by diflubenzuron in zebrafish embryos from five aspects. I suggest that it can be accepted after  major  revision.

1. In 1. Instroduction, the statment"Diflubenzuron helps agricultural industry improve the production" does not make sense.

2.  Figure 1. (A), the structure of Diflubenzuron should be rewrited, giving the correct bond angles.

3. In "2. Results", some statement is not easy to understand, such as “Diflubenzuron exposure inhibited the hatchability and average beats of heart (Fig. 1C and 1D); besides, it in-duced the pericardial edema of embryos”, etc, should be polished.  

4. In "3. Discussion",in  the sentence "for controlling agricultural pest [19]. Thus, indis-criminate use of diflubenzuron often causes the residue in nvironment." "Thus" is not sutible. In the second paragraph, why did you mention "deltamethrin and pyridaben" , you should connect them with your results.

Author Response

The manuscript study the cardiotoxicity induced by diflubenzuron in zebrafish embryos from five aspects. I suggest that it can be accepted after major revision.

  1. In 1. Instroduction, the statment"Diflubenzuron helps agricultural industry improve the production" does not make sense.

Response: Thank you. We have removed this sentence.

  1. Figure 1. (A), the structure of Diflubenzuron should be rewrited, giving the correct bond angles.

Response: Thank you. According to the suggestion of reviewer 1, fig 1A is necessary for the Ms, so, we have deleted it.

  1. In "2. Results", some statement is not easy to understand, such as “Diflubenzuron exposure inhibited the hatchability and average beats of heart (Fig. 1C and 1D); besides, it in-duced the pericardial edema of embryos”, etc, should be polished.  

Response: Thank you. We have polished these sentences in lines of 84-85.

  1. In "3. Discussion",in  the sentence "for controlling agricultural pest [19]. Thus, indis-criminate use of diflubenzuron often causes the residue in nvironment." "Thus" is not sutible. In the second paragraph, why did you mention "deltamethrin and pyridaben" , you should connect them with your results.

Response: Thank you. We have removed it. These examples are added to explain the rationality of the high and low half lethal concentration.

Round 2

Reviewer 3 Report (New Reviewer)

The manuscript has been revised, and I think it is suitable for publication.

This manuscript is a resubmission of an earlier submission. The following is a list of the peer review reports and author responses from that submission.

Round 1

Reviewer 1 Report

I would like to highlight the relevance of the work carried out, for studying the cardiotoxicity of a pesticide with an excellent experimental design, in my opinion, the first time studying this insecticide in cardiotixicity in zebrafish as a method. The work can be improved by adding the introduction if the pesticide, which is widely used, causes toxicity in humans in addition to aquatic animals. In the material and methods, explain the choice of doses used. And in the discussion: knowing the genetic similarity of zebrafish with humans, discuss the correlation of the findings between the species.

Author Response

Response: Thank you for your comments and suggestions.

  1. diflubenzuron also induces toxicity to human, the relevant information was added in lines of 52-54 (Wang et al., 2002, Analytica Chimica Acta, 468:209-215).
  2. The preliminary experiment showed the no-toxic phenotype in zebrafish embryos when the concentration of diflubenzuron was lower than 0.5 mg/L, however higher concentration of diflubenzuron (exceed of 3.5 mg/L) caused over 60% mortality at 96 hpf. Therefore, the concentrations of 0.5-2.5 mg/L were used in this study. This explanation was added in lines of 226-229.
  3. There are about 30,000 genes in zebrafish genome, which are similar to human genes, with the conservation of 87% ((Reimers et al., 2004, The Journal of biological chemistry,279:38303-38312; Zetterberg et al., 2006, J. Biol. Chem, 281:11933-11939). Therefore, many studies have used zebrafish as a model to reveal the mechansim of human disease, drug screening and toxicology evaluation, etc ((Lieschke and Currie, 2007, Nat. Rev. Genet. 8:353-367; Zon and Peterson, 2005, Nat. Rev. Drug Discov. 4:35-44). The relevant information was added in lines of 59-62.

Reviewer 2 Report

The paper by Xue Han et al. investigated the cardiotoxic effects of Diflubenzuron (DFB), an insecticide, on zebrafish embryos. Authors show that DBF significantly decreased survival, hatchability, and heart rate in embryos at 6 h to 96 h post-fertilization (hpf). Furthermore, DBF induced cardiac edema, significantly impaired the expression of major genes involved in cardiac development (nppa, gata4, vmhc, tbx5, myh6, tbx2b, bmp4 and klf2a) and increased oxidative stress (increased reactive oxygen species production and reduced expression of anti-oxidant enzymes). Finally, authors show increased expression of apoptotic genes and conclude that DFB leads to cardiotoxicity in zebrafish, therefore recommending caution with the use of this insecticide.

The lack of previous studies on the potential toxicity of DFB in aquatic organisms makes this topic relevant. Nevertheless, major issues weaken the conclusion of the study.

Major points:

-          The study is highly descriptive and does not provide any speculation on the possible mechanism behind the cardiotoxicity of DFB.

-          Authors fail to identify a non-effect dose of the compound. Most of the functional parameters analyzed as well as the expression of most of the genes are already profoundly altered by the lowest dose employed (0.5 mg/ml).

-          The lack of comparative analysis of DBF cardiotoxicity with that of other insecticides further weakens the conclusion of the study.

-          There is lack of clarity in the methods. All experiments are realized on n=24. Does this n refer to independent experiments? I could not find this information in the text. What is the method used to measure beats/min (manually, automatically?)? In the methods section, authors specified that the experiment was performed at 96hpf, while in all the figures it seems that experiments were realized at 72hpf.

-          Fig1B. Authors calculated the LC50. However, for a correct estimation, doses between 2.5mg/L and 4.5mg/L should be included. Moreover, the LC50 should be referred on the graph and the logarithmic curve should be provided. The timing of the experiment is not specified.

-          Fig 1C-D-E. The effects are not evaluated at the LC50.

-          Authors only focus their attention on the cardiotoxicity of DBF, but systemic toxicity cannot be excluded. Systemic toxicity parameters, like body length and body shape should be assessed as well. Furthermore, additional cardiotoxicity parameters, like cardiac looping, fractional shortening and red blood cells velocity should be evaluated.

-          Fig3. Nppa expression is usually shown along with that of nppb to demonstrate cardiac damage. Moreover, these markers are representative of cardiac injury. How do authors explain the finding that nppa expression is decreased after DFB exposure?

-          Fig5. Cell death is evident starting from 2.5mg/L, but this observation is not in line with expression of the apoptotic genes represented in the other graphs. How can authors explain this discrepancy? TUNEL assay could be performed to strengthen this result.

Minor point:

-          The layout of the paper should be revised. The figures are lacking uniformity

-          English and the syntax should be revised. The text lacks fluidity and links between paragraphs.   

-          Error of annotation from the figure 5.

-           “SOD” and “CAT”  abbreviations are not specified in the text.

-          Was DFB directly added to the embryo’s water?

Author Response

Major points:

  1. The study is highly descriptive and does not provide any speculation on the possible mechanism behind the cardiotoxicity of DFB.

Response: Thank you. We speculated that the developmental toxicity and neurotoxicity induced by diflubenzuron might be caused by ROS-induced apoptosis in the process of proliferation. This information was added in lines of 199-201.

  1. Authors fail to identify a non-effect dose of the compound. Most of the functional parameters analyzed as well as the expression of most of the genes are already profoundly altered by the lowest dose employed (0.5 mg/ml).

Response: Thank you. In our preliminary experiment, the embryos were exposed to 0.1-0.4 mg/L of diflubenzuron at 96 hpf. There are no-toxic phenotypes from 0.1-0.4 mg/L of diflubenzuron-exposed embryos and also the expression of cardiac development-related genes have no obvious changes.

  1. The lack of comparative analysis of DBF cardiotoxicity with that of other insecticides further weakens the conclusion of the study.

Response: Thanks for your suggestion. We have added two examples about insecticides-induced cardiotoxicity. The relevant information was added in lines of 134-135 ( Li et al., 2019,Chemosphere, 219:155-164; Ma et al., 2021, Aquatic Toxicology, 237:105870).

  1. There is lack of clarity in the methods. All experiments are realized on n=24. Does this n refer to independent experiments? I could not find this information in the text. What is the method used to measure beats/min (manually, automatically?)? In the methods section, authors specified that the experiment was performed at 96hpf, while in all the figures it seems that experiments were realized at 72hpf.

Response: Thank you. “n=24” means the number of embryos in each experiment. Our results were showed by three independent experiments. The clear presentations were added in each figure legend. We manually measure heart rates (beats/min) through timer with stereomicroscope. The embryonic heart of zebrafish starts to develop from 5 hpf, and it takes only 72 hours to mature and function (Bakkers, 2011, Cardiovasc, 91:279-288; Zennaro et al., 2014, PLoS Clinical Trials.,9: e98131). Therefore, we selected 72 hpf as the monitoring endpoint. We revised the related-description.

  1. Fig1B. Authors calculated the LC50. However, for a correct estimation, doses between 2.5mg/L and 4.5mg/L should be included. Moreover, the LC50 should be referred on the graph and the logarithmic curve should be provided. The timing of the experiment is not specified.

Response: Thank you. We provided the LC50s at 72 hpf and 96 hpf in results and referred them on the graph.

  1. Fig 1C-D-E. The effects are not evaluated at the LC50.

Response: Thank you. The LC50 of diflubenzuron always causes many death and malformation. However, the aim of this study is to uncover the toxicity mechanism caused by diflubenzuron. Therefore, we chose the concentrations lower than LC50.

  1. Authors only focus their attention on the cardiotoxicity of DBF, but systemic toxicity cannot be excluded. Systemic toxicity parameters, like body length and body shape should be assessed as well. Furthermore, additional cardiotoxicity parameters, like cardiac looping, fractional shortening and red blood cells velocity should be evaluated.

Response: Thanks for your suggestions. Diflubenzuron exposure may induce systemic toxicity. However, in this study, we did not find the changes of body length, but pericardial edema of embryos were induced after diflubenzuron exposure. Our data in the development of cardiac, the expression of cardiac development-related genes and apoptosis of cardiac cells can preliminarily confirm the cardiotoxicity of diflubenzuron. Your suggestions about cardiotoxicity parameters will be accepted in our future study.

  1. Fig3. Nppa expression is usually shown along with that of nppb to demonstrate cardiac damage. Moreover, these markers are representative of cardiac injury. How do authors explain the finding that nppa expression is decreased after DFB exposure?

Response: Thank you. During the whole embryonic and fetal development, the decrease of nppa expression may indicate that the atrial and ventricular functions are affected in the early cardiac phenotype stage of embryonic development. (Houweling et al., 2005, Cardiovasc Res, 67:583-593). The relevant information was added in lines of 158-160.

  1. Fig5. Cell death is evident starting from 2.5mg/L, but this observation is not in line with expression of the apoptotic genes represented in the other graphs. How can authors explain this discrepancy? TUNEL assay could be performed to strengthen this result.

Response: Thank you. The occurrence of apoptosis is always not completely incosistent with the changes of genes expression. However, in view of your concern, we re-conducted this experiment. The weak apoptosis can be found in groups of 0.5 mg/L and 1.5 mg/L.

Minor point:

  1. The layout of the paper should be revised. The figures are lacking uniformity

Response: Thank you. The layout of the paper has been improved.

  1. English and the syntax should be revised. The text lacks fluidity and links between paragraphs.   

Response: Thank you. The manuscript has been polished considerably, including of spellings, grammars and structures.

  1. Error of annotation from the figure 5.

Response: Thank you. We have revised it.

  1. “SOD” and “CAT” abbreviations are not specified in the text.

Response: Thank you. SOD is superoxide dismutase, CAT is catalase. The abbreviations were added in lines 109-110.

  1. Was DFB directly added to the embryo’s water?

Response: Thank you. Firstly, we dissolved the diflubenzuron powder in dimethyl sulfoxid (1 mg/L). Then, the different dose of diflubenzuron solution was added to embryo’s water.

Round 2

Reviewer 2 Report

Although I appreciate the efforts of authors to discuss in the main text some of the points that this reviewer has raised in the previous report, the manuscript did not show any significant improvement with respect to the content compared to the previous version.